# Investigation of Pretreatment of Textile Wastewater for Membrane Processes and Reuse for Washing Dyeing Machines

**DOI:** 10.3390/membranes12050449

**Published:** 2022-04-21

**Authors:** Iva Ćurić, Davor Dolar

**Affiliations:** Department of Physical Chemistry, Faculty of Chemical Engineering and Technology, University of Zagreb, Marulićev Trg 19, 10000 Zagreb, Croatia; dolar@fkit.hr

**Keywords:** membrane technology, sand filtration, textile wastewater, ultrafiltration, reuse

## Abstract

The aim of this study was to investigate the best pretreatment of textile wastewater (TWW) for membrane separation processes and the previously unexplored reuse of treated TWW for washing dyeing machines. Sand filtration (SF), coagulation, coagulation/flocculation, and ultrafiltration (UF) with hollow fiber membrane (ZW1) were used for pretreatment. Pretreatment selection was based on turbidity, total organic carbon (TOC), and color. SF and ZW1 were found to be the best pretreatments. In addition, the SF and ZW1 effluents were subjected to the 5 (PT) and 50 (MW) kDa UF flat sheet membranes to test removal efficiency. ZW1-PT was better in terms of removal results and fouling. To reduce the use of drinking water for washing dyeing machines, the characteristics of ZW1-PT effluent were compared with drinking water from a textile factory. TWW treated with this hybrid process fulfils the purpose of reuse for washing dyeing machines and can be used in Galeb d.d., Croatia, or in any other textile factory, saving up to 26,000 m^3^ of drinking water per year. This contributes to both sustainable production and the conservation of water resources.

## 1. Introduction

The textile industry, one of the largest industries in the world, generates a significant amount of wastewater through production processes such as dyeing, washing, and bleaching [1,2]. These processes require dyes, auxiliaries, finishing chemicals, and large amounts of water [3,4]. For this reason, textile wastewater (TWW) has high biochemical oxygen demand (BOD), chemical oxygen demand (COD), turbidity, total organic carbon (TOC), conductivity (*κ*), and color [5,6]. Various processes have been used for the treatment and reuse of TWW in the textile industry, for example, physico-chemical methods such as chlorination, coagulation, coagulation/flocculation, adsorption, and advanced oxidation processes (ozonation and Fenton treatment) [7,8,9]. Each of these processes has some serious disadvantages in terms of sludge generation and is not economically justifiable [10].

The use of membrane separation processes (MSPs) has become widely accepted in the removal of pollutants and reuse of water. These processes help the industry to close the water loop and reduce freshwater consumption. Kamali et al. [6] evaluated and ranked MSPs as the most sustainable technology for industrial wastewater treatment according to technical, environmental, economic, and social aspects using the Fuzzy–Delphi approach.

Over the years, several studies have focused on the treatment of TWW with MSPs, but also on the pretreatment (e.g., sand filtration (SF), coagulation, combination of coagulation/flocculation) of TWW. These pretreatments were used to prevent membrane fouling and to achieve higher removal efficiency [11,12,13,14,15,16].

Wastewater from textile industry can be reused in various ways, namely: use for wet processes, e.g., dyeing, washing, bleaching; use as cooling water, for washing stages in wet processes that do not require high quality treated water; etc. [17]. The reuse of treated TWW for wet processes has already been studied in detail [16,18,19], but as far as the authors are aware, there is no study on the reuse of treated TWW for washing dyeing machines. Dyeing machines must be washed after each dyeing process to prevent the previous dyes from being transferred to the new dyes.

Therefore, this study presents a comprehensive investigation of the best TWW pretreatments (coagulation, coagulation/flocculation, SF, and a UF hollow fiber membrane) for the membrane separation process to ensure the most economical hybrid process based on physico-chemical analysis and fouling monitoring. The treated effluent was analyzed for the possibility of reuse in washing dyeing machines to reduce the consumption of natural water resources.

## 2. Materials and Methods

### 2.1. Sampling and Analytical Methods

The untreated TWW samples required for this study were taken from the discharge effluent of a textile factory (Galeb d.d.) in Omiš, southern Croatia (Dalmatia), which specializes in the production of clothing/apparel of the “cotton” (CO) category. Samples were taken from an equalization tank and were collected in 25 L plastic containers. They were stored at a temperature below 10 °C.

Analysis of all samples was performed according to Standard methods [20]. TOC was determined using the Shimadzu TOC-V_ws_ carbon analyzer (Kyoto, Japan), and turbidity was measured using the WTW Turb 430 turbidimeter (Weilheim, Germany). Conductivity and pH were measured using the SI Analytics HandyLab 680 (Mainz, Germany). Anions and cations were analyzed using an Ion chromatograph DIONEX IC-3000 (Thermo Fisher Scientific, Sunnyvale, CA, USA). Color was determined using a Hach Lange DR3900 spectrophotometer (Berlin, Germany) and was expressed by the Spectral absorption coefficient (SAC), which was determined based on absorbance measurements using the spectrophotometric method at three wavelengths (*λ* = 436, 525, and 620 nm) according to DIN-38404/1 and Equation (1):(1)$$SAC=\frac{1000\times E\left(\lambda \right)}{d}\;\left(m^{-1}\right)$$
where *E*(*λ*) is the absorbance at a given wavelength *λ*, and *d* is the thickness of the measurement cuvette (mm). The optimal concentration of coagulant and flocculant and pH were determined using response surface methodology (RSM).

Total permanent water hardness is expressed according to Equation (1) [21]:[CaCO_3_] = 2.5 × [Ca^2+^] + 4.1 × [Mg^2+^]      (mg L^−1^)(2)

To characterize the fouling layer, the surface of the pristine and fouled membranes were analyzed using Fourier transform infrared (FTIR) spectroscopy of the Bruker Vertex 70 equipped with a Platinum ATR single reflection diamond (*n* = 2.4) crystal-based module in the mid IR range (400–4000 cm^−1^). The membrane spectra were recorded with a resolution of 4 cm^−1^ and 32 scans.

### 2.2. Lab Scale Experimental Procedures

To obtain the best possible pretreatment for MSPs, four pretreatment processes were selected and investigated (sand filtration, coagulation, coagulation/flocculation, and UF hollow fiber). For coagulation, 40% FeCl_3_ was used as the coagulant, and the experiment was performed with 1 L TWW, using 12 different concentrations of coagulant ranging from 0.35 to 4.48 mM FeCl_3_ at different pH values (3.5, 4.5, 5.5, 7.3, and 8.5). Each supernatant was collected, and turbidity, TOC, and color were determined. The coagulation/flocculation process included the addition of the optimal concentration (2.07 mM) of coagulant (FeCl_3_) at a pH of 5.95 and six concentrations of MagnaFloc LT25 anionic flocculant (0.1–1.0 mg L^−1^). In addition, each supernatant was collected and analyzed as in the previous step. The optimal conditions for coagulation were calculated using the coefficient of determination or regression coefficient R-squared (R^2^) and analysis of variance (ANOVA). SF was performed through a column (55 cm high with a diameter of 5.5 cm) filled with quartz filter sand Kema, Baumit Croatia (grain size ranged from 0.18 to 1.85 mm). UF with hollow fiber ZW1 module of 200 kDa (GE Water and Process Technologies, Oroszlany, Hungary) was performed in a laboratory setup (described in detail in Dolar et al. [22]) operating in continuous mode at a transmembrane pressure of about −0.35 bar, a permeate flux of 18 L m^−2^ h^−1^, and an air supply rate of 15 L min^−1^. In both cases, filtrate and permeate were collected and analyzed.

After the two best pretreatments were determined, the effluents were subjected to UF membrane treatment. Two commercially available UF flat sheet membranes, 5 and 50 kDa (PT and MW), were provided by GE Water & Process Technologies (Delfgauw, The Netherlands). The main characteristics of these membranes (material, MWCO, maximum working pressure, pH range, and typical flux) are listed in Table 1. Before use, the membranes were stored in a cold and dark place (refrigerator).

UF experiments were performed in batch circulation mode, i.e., permeate and retentate were returned to the feed tank (10 L) in a crossflow Sepa CF II cell (Sterlitech Corporation, Auburn, AL, USA) with a membrane area of 138 cm^2^, a spacer of 0.45 mm, and channel dimensions of 14.5 × 9.5 × 0.17 cm^3^. Details of the device can be found in Racar et al. [23]. The feed was pumped through the cell at a rate of 3 L min^−1^, with a cross-flow velocity of 0.75 m s^−1^. The working pressures for the PT and MW membranes were 4 and 1.5 bar, respectively, and were reported by the manufacturer as typical working pressures.

First, the membranes were washed with demineralized water (3 L) without pressure to wash out the conserving agents. Then, the membranes were flux stabilized with demineralized water at the appropriate working pressure for each membrane. Once the flux was stabilized, TWW was added and circulated in the system for 2 h. Flux was monitored using a technical balance KERN 440-35A (Balingen, Germany) while the computer recorded mass every 10 s. After treatment, permeate samples were taken to analyze the monitored parameters. At the end, the membranes were washed with demineralized water at working pressure. Finally, the washed membranes were dried in a dryer at 35 °C for about 15 h to remove residual water. The membranes dried in this way were analyzed by FTIR.

## 3. Results

### 3.1. Characterization of Textile Wastewater

For this study, the TOC concentration, turbidity, and color of the sampled TWW (see Figure 1 and Figure 2) were determined. They were within the limits published in a previous study by Yaseen et al. [24] for typical textile effluents from different sources and countries. Real TWW samples were collected from three batches. Each batch was homogenized and characterized for each pretreatment experiment. The differences observed between samples are due to (1) changes in the real wastewater over time and (2) homogenization of the samples. The presence of color is related to the use of organic dyes in production processes, which cause high TOC concentrations due to their poor degradability [25]. Since suspended solids are present in TWW, this leads to turbidity [26].

### 3.2. Removal of Pollutants in the Examined Pretreatments

Figure 1 and Figure 2 show the results of turbidity, TOC, and SAC reduction with their removal efficiencies for each of the pretreatment processes studied.

#### 3.2.1. Sand Filtration

To reduce the turbidity, TOC, and color of TWW, the physico-chemical process SF was chosen, which is characterized by simplicity, low operating cost, and efficiency, and is most commonly used for wastewater pretreatment or as a tertiary treatment [27]. SF reduced turbidity from the initial 140 NTU in the TWW to 82 NTU in the analyzed filtrate, corresponding to 41.4%. In addition, the TOC values showed a slightly lower removal from the initial value of 363.4 mg L^−1^ in the influent sample of the TWW to 242.3 mg L^−1^ in the filtrate, corresponding to a removal efficiency of 33.3%. Lindroos et al. [28] also showed in their study that the sand filter successfully reduced TOC, possibly due to the transport of O_2_ and nutrients into the bio-layer. The sand filter was efficient in removing particles, solutes, and organic matter from the wastewater. This is in agreement with the study of Verma et al. [29]. The color removal results showed a slight increase of 0.02 m^−1^ at 436 nm and 0.04 m^−1^ at 525 and 620 nm. This may be attributed to the dissolution of some sand grains and probable further coloration of the filtrate by the chemicals present in the TWW.

#### 3.2.2. Coagulation

The coagulation process was chosen as one of the pretreatments because it is the most commonly used and effective method for treating dyes in wastewater. After conducting a series of jar tests with 40% FeCl_3_ at different pH and coagulant concentrations, an analysis of the supernatant was performed, and the optimum conditions were calculated using RSM. The results of the jar experiments are shown in Appendix A, while the resulting response surfaces for turbidity, TOC, and color are shown in Appendix A. The results show an optimal pH of 5.95 and a coagulant concentration of 2.24 mM. Under these conditions, the removal efficiencies for turbidity, TOC, SAC_436 nm_, SAC_525 nm_, and SAC_620nm_ were 86.1%, 40.8%, 81.9%, 85.3%, and 83.3%, respectively. The coagulation process showed high removal efficiency under optimal conditions, which is consistent with Naghan et al. [30] in that pH is an important factor for maximum removal efficiency. Dalvand et al. [31] confirmed high color removal under optimal conditions with FeCl_3_. A turbidity, color, and TOC removal rate of more than 70% with FeCl_3_ was also confirmed in the study by Lara et al. [32].

#### 3.2.3. Coagulation/Flocculation

The use of a coagulation/flocculation treatment is widely used for dye removal and is usually used as a pretreatment for some of the MSPs [33]. The results of the Jar experiments are shown in Appendix A, although the resulting response surfaces could not be calculated due to the same pH (5.95) defined as the optimal value for coagulation with FeCl_3_. The jar test showed the highest turbidity removal from 202 to 6.4 NTU (96.8%) and TOC removal from 528.3 to 265.4 mg L^−1^ (49.8%) with 0.2 g L^−1^ MagnaFloc LT25. Results from SAC showed 653 m^−1^ at 436 nm in the feed sample and a decrease to 49 m^−1^ (92.5%) in the supernatant, a decrease from 436 to 29 m^−1^ (93.3%) at 525 nm, and from 313 to 38 m^−1^ (87.9%) at 620 nm. The removal efficiency in the case of coagulation/flocculation is higher than in the case of coagulation. The flocculation process contributed to a better removal of these parameters, but the addition of another chemical and the precipitation of a larger amount of sludge. In our experiments, up to 500 mL of sludge was produced, which increases the total operating cost of TWW treatment. Therefore, the coagulation/flocculation process was not investigated further.

#### 3.2.4. UF Hollow Fiber ZW1

UF hollow fiber membranes are used for TWW treatment because they have excellent (up to 99%) color decrease [34]. Such membranes can successfully reduce both fouling and concentration polarization during subsequent treatment with MSPs. Therefore, the efficiency of UF hollow fiber membranes as TWW pretreatment was also investigated in this study. The results showed a decrease in turbidity from 196 in the feed to 12.4 NTU in the permeate and in TOC from the initial 322.6 to 229.0 mg L^−1^. In addition, a significant decrease in SAC was observed at 436 nm from 98 to 19 m^−1^, at 525 nm from 52 to 5 m^−1^, and at 620 nm from 40 to 3 m^−1^. This is consistent with the fact that the UF hollow fiber membrane removes the colloidal material that causes turbidity [35]. The relatively low TOC removal efficiency indicates that textile wastewater contains low molecular weight organic matter [36]. Alardhi et al. [37] confirmed the success of dye removal with UF hollow fiber membranes, and Sherhan et al. [38] also indicate high removal of turbidity of 95% and TOC of 96%.

### 3.3. Effects of SF and ZW1 Effluents on 5 and 50 kDa Membranes Treatment

To test pretreatment for MSPs, two pretreatments (SF and ZW1) were selected. Their effluents were additionally treated with two UF flat sheet membranes, 5 kDa PT and 50 kDa MW. The results are included in Table 2. It can be seen that there is a difference between the influent samples (SF filtrate and ZW1 permeate). This could be due to insufficient homogenization before/during treatment and differences in the TWW sample affecting the removal efficiency. The permeates of SF-PT and SF-MW showed removal efficiencies of turbidity, TOC, and SAC_436, 525, 620 nm_ of 99.1–97.5%, 90.0–28.1%, and 98.1–95.1%, respectively, while those of ZW1-PT and ZW1-MW showed removal efficiencies of 5.3–86.2%, 94.1–54.8%, and 69.4–66.7%, respectively. The PT membrane showed higher removal efficiency in both pretreatments. Membranes with higher MWCO (MW) showed lower removal efficiencies, which is confirmed by Ćurić et al. [39].

### 3.4. Effect of Pretreatment on the UF Flat Sheet Membrane Flux

To test a good pretreatment for MSPs, membrane fouling needs to be studied. Therefore, fouling was monitored in this study using the normalized flux decrease. The normalized fluxes (*J*/*J*_0__(water)_) during the 2 h treatment for the combination of SF and ZW1 with PT and MW are shown in Figure 3. The decrease in flux is explained by the initial decrease (comparing the flux of demineralized water (*J*_0(water)_) with the first flux of the treatment) and by the decrease in flux during the 2 h treatment. The initial decrease for ZW1-PT, SF-PT, ZW1-MW, and SF-MW was 20%, 25%, 25%, and 50%, respectively. In all cases shown, the initial decrease in pretreatment ZW1 was smallest for the membrane PT, followed by a similar decrease for the combination ZW1-MW and SF-PT. The largest difference is visible in the normalized flux decrease during the 2 h treatment. Again, the decrease was lowest for ZW1-PT (20%), whereas it was 48%, 52%, and 37% for SF-PT, ZW1-MW, and SF-MW, respectively. With the exception of the lowest initial normalized flux for the ZW1-PT combination, the flux decline was relatively linear during treatment, suggesting that the flux decline may be due to concentration polarization.

These values can be related to the fact that the MWCO of the membrane strongly contributes to flux drop, but also to retention, as can be seen from the above results in Table 2. Both PT membranes showed lower flux drop regardless of pretreatment. The results are in agreement with the study of Ćurić et al. [39], which showed that a cake layer forms in the membrane with the highest MWCO, in this case MW, at the beginning of the treatment as a result of rapid pore blocking and adsorption. Treatment of ZW1 permeate showed better results for normalized flux, as the pretreatment resulted in higher overall removal efficiency and lower fouling on the UF flat sheet membranes. As can be seen, this is not the case for the SF pretreatment. The decrease in normalized flux for ZW1-MW is due to residual foulants in the ZW1 permeate that may stick to the membrane surface or in the pores.

FTIR spectra (Figure 4b–d) confirmed fouling by formation of new peaks and even disappearance of peaks compared to pristine membranes. ZW1-PT showed a small peak at 3500 cm^−1^, while ZW1-MW showed a new peak at 3367 cm^−1^ and disappearance of peaks at 550 and 2500 cm^−1^. SF-PT had three new peaks at 2849, 2918, and 3303 cm^−1^. SF-MW showed four new peaks at 714, 806, 2304, and 2859 cm^−1^. The disappearance of peaks during treatment can be explained by the degradation of organic compounds and the formation of new compounds as organic intermediates. In the FTIR spectra (fouled and pristine membranes), the peak is seen at 1500 cm^−1^. This stretching represents the azo group (C=N), the chromophore group of the dyes [40]. This indicates that the dye was not completely removed from the TWW, as shown in Table 2. The probable functional groups of the fouled and pristine membranes are summarized in Table 3.

### 3.5. Selecting the Best Pretreatment

When the pretreatments were tested individually (Figure 1 and Figure 2), the ZW1 hollow fiber membrane proved to be better than SF, as removal efficiencies were higher. When the effluents of SF and ZW1 were tested on UF flat sheet membranes (5 and 50 kDa), the opposite situation was observed, i.e., SF showed higher removal efficiencies compared to ZW1. One of the reasons for this could be the different concentrations of the feed. From the results presented previously (Table 2), the concentration of the feed was lower in the second part of the study, i.e., when testing the effluents of SF and ZW1 on 5 and 50 kDa membranes. This was most evident in the turbidity and color when treating ZW1-PT and ZW1-MW. These differences, also explained in Section 3.1., were likely caused by changes in real TWW as the samples awaited experiments. However, when the normalized flux drop for the UF flat sheet membrane treatment was examined, it was found that ZW1 (20% and 25%) had a lower normalized flux drop at the beginning of the treatment than SF (25% and 50%). When treating SF filtrate, it is desirable to clean the membrane immediately, which would ultimately mean higher processing costs since chemical cleaning is included in overall costs. It is common to perform chemical cleaning when the normalized parameters, e.g., normalized flux, have decreased from 10% to 20%. Therefore, it is clear that chemical cleaning must be performed after 2 h for the ZW1-PT combination, since the normalized flux has decreased by 20% during the treatment. For the other combinations, chemical cleaning must be performed after only a few minutes of treatment. It can be concluded that the cost of pretreatment with the ZW1 is the lowest.

### 3.6. Possibility of Reusing TWW in Washing Dyeing Machines

In order to protect the environment and reduce the consumption of drinking water for washing dyeing machines, the possibility of reusing TWW is investigated using the Galeb d.d. factory, from which the TWW samples were taken. All data were taken from Galeb d.d.

In order to better represent the water consumption for washing dyeing machines, a schematic representation (Figure 5) of the process, which consists of three stages, was made as follows:

The auxiliaries Losin OCB-O (Textilcolor AG) (2 g L^−1^), TECOTEX FTK (Textilcolor AG) (2 g L^−1^), NaOH (Velekem d.d.) (5 g L^−1^), sodium-hydrosulfite (Velekem d.d.) (2 g L^−1^), and sodium-hypochlorite (Velekem d.d.) (3 g L^−1^) were added to 3 m^3^ of tap water at 40 °C in the dyeing machine. The temperature was increased to 98 °C and circulated for 30 min. The liquid was drained off.

In the second step, new water was added at 98 °C without auxiliary and circulated for 10 min.

In the last step (cold wash), water at a temperature of 20 °C was added and circulated for 5 min.

Each stage of the washing dyeing machines shown in Figure 5 required three changes of water.

The amount of water required depends on the capacity of the machines. The factory has two machines with a capacity of 100 kg each, one machine with 500 kg, and one machine with 300 kg. When the factory is at full capacity, all machines are occupied per shift, and the machines must be washed after each dyeing process. This would mean that a total of 30 m^3^ of water is used per shift to wash the machines. Since the factory works two shifts on weekdays from Monday to Friday, not counting one month of collective annual leave and about 10 vacation days, the water consumption for washing the dyeing machines is about 26,000 m^3^ per year.

In order to save this amount of drinking water, the effluent from the selected hybrid process ZW1-PT was analyzed physico-chemically and compared with the drinking water of the factory, which uses this water to wash the dyeing machines. Table 4 shows the parameters of the drinking water used in the production process and the ZW1-PT permeate. This permeate had higher values for conductivity (71%) and pH. Conductivity is not a problem for washing machines, as large amounts of salts are added during the dyeing process, causing high conductivity. In the case of pH, it is important to mention that pH during the dyeing process is higher than 9 and the pH of ZW1-PT is still in the range of drinking water according to the Croatian legislation (6.5–9.5). Other tested parameters gave better results than the drinking water, especially calcium and magnesium salts, which cause water hardness and consequently scale formation on the machine pipes and cracking in these pipes. From the comparative results (Table 4), it can be concluded that the permeate ZW1-PT can be used for washing dyeing machines in the textile industry, which can significantly reduce the consumption of drinking water and thus protect the environment.

## 4. Conclusions

In this study, pretreatments of real TWW were performed for treatment with MSPs. SF, coagulation, coagulation/flocculation, and UF hollow fiber (ZW1) pretreatments were evaluated for turbidity, TOC, and color removal. The coagulation and coagulation/flocculation processes were effective in removing the studied parameters, but the high sludge production and addition of chemicals make these treatments expensive. SF had lower removal results than ZW1. These two pretreatments were selected, and their effluents were tested with 5 and 50 kDa UF flat sheet membranes. The results showed that the ZW1 permeate had slightly worse removal results than the SF filtrate. However, further investigation of the normalized flux showed a lower flux drop when treating ZW1 permeate compared to a higher drop when treating SF filtrate. Thus, it was shown that hybrid treatment ZW1-PT can reduce the operating cost of TWW treatment and also provide a permeate suitable for washing dyeing machines after dyeing processes.

## Figures and Tables

**Figure 1 membranes-12-00449-f001:**
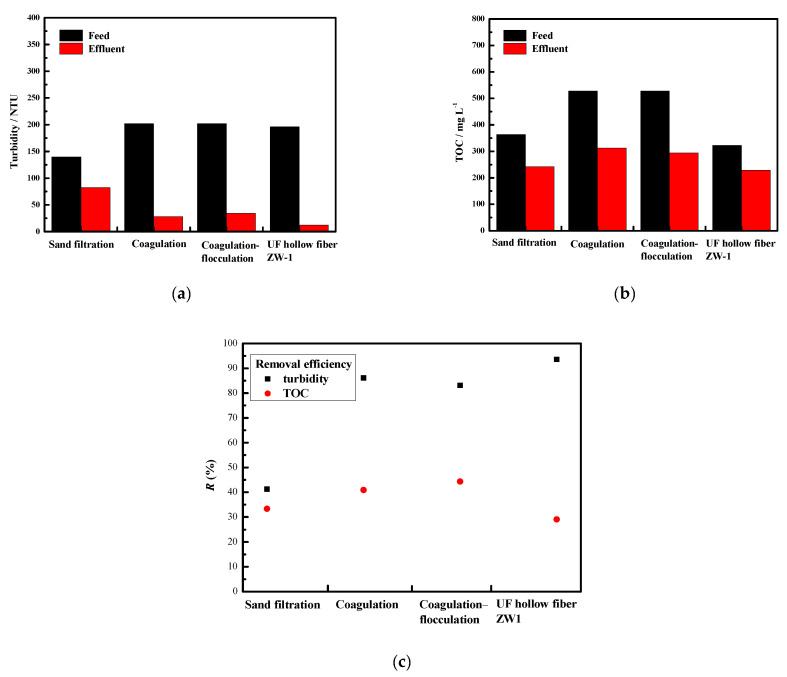
Concentration of (**a**) turbidity and (**b**) TOC in real TWW (feed) and investigated pretreatment effluents with (**c**) removal efficiency (*R*).

**Figure 2 membranes-12-00449-f002:**
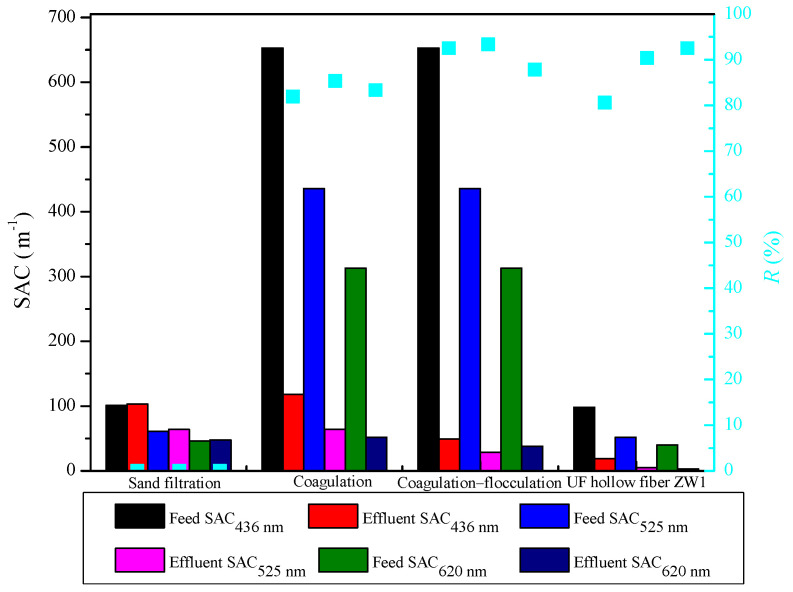
Color (SAC) of real TWW (feed) and investigated pretreatment effluents with removal efficiency (*R*).

**Figure 3 membranes-12-00449-f003:**
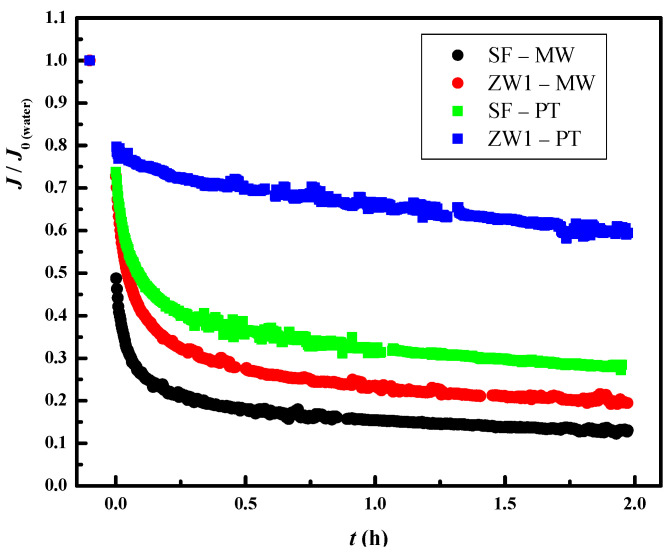
Normalized fluxes (*J*/*J*_0(water)_) of SF and ZW1 pretreatments with PT and MW membranes.

**Figure 4 membranes-12-00449-f004:**
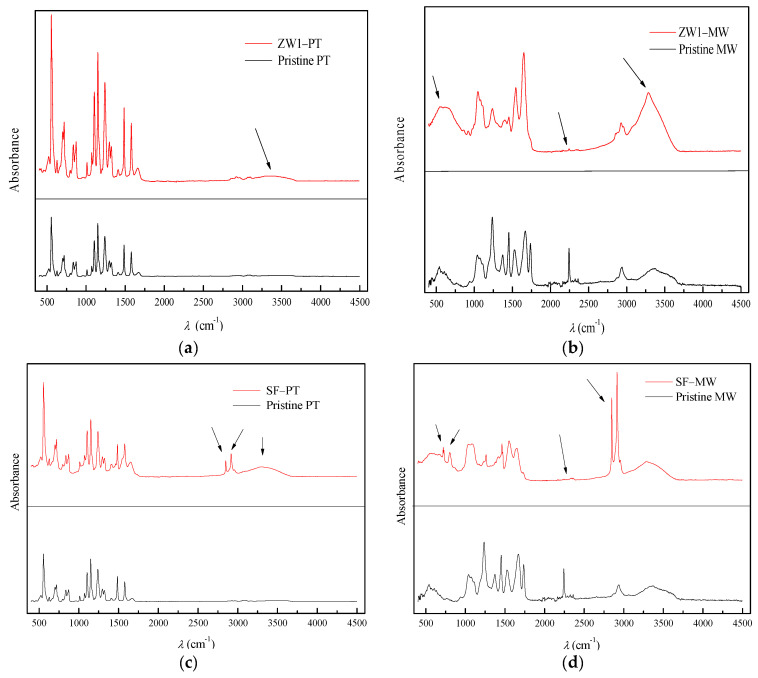
FTIR spectra of pristine and membranes after treatment: (**a**) ZW1-PT, (**b**) ZW1-MW, (**c**) SF-PT, and (**d**) SF-MW.

**Figure 5 membranes-12-00449-f005:**
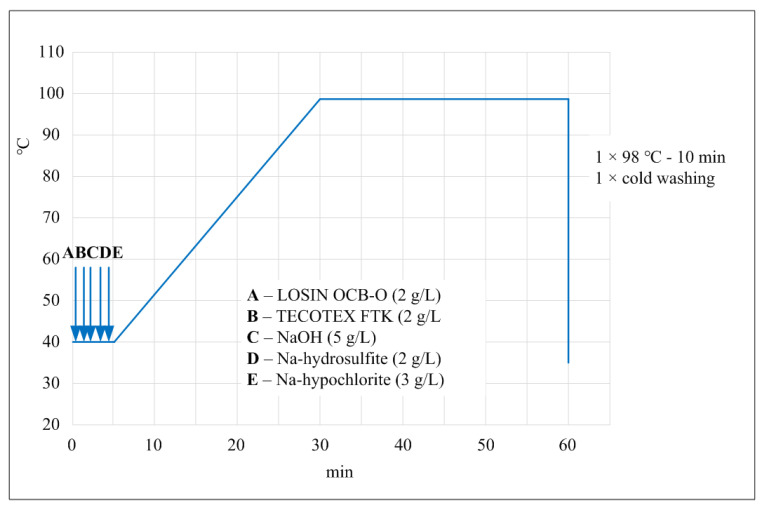
Schematic representation of the washing dyeing machines.

**Table 1 membranes-12-00449-t001:** The main characteristics of used membrane for UF treatment.

Membrane	Material	Maximum Working Pressure/Bar	pH Range	Typical Flux/Pressure LMH/Bar	MWCO/kDa
PT	PES/PSf ^1^	10	1–11	153/3.45	5
MW	PAN ^2^	7	2–9	299/1.32	50

^1^—Poly(ether-sulfon)/polysulfone; ^2^—Polyacrylonitrile.

**Table 2 membranes-12-00449-t002:** Physico-chemical characteristics of SF-PT/MW and ZW1-PT/MW combination.

Parameter		Sample	SF-PT	SF-MW	ZW1-PT	ZW1-MW
Turbidity (NTU)		Influent	90.1	61.9	3.57	23.2
		Effluent	0.85	1.55	3.38	3.19
TOC (mg L^−1^)		Influent	195.8	266.9	311.0	157.8
		Effluent	19.5	192	18.2	71.3
Color (SAC)	436 nm (m^−1^)	Influent	160	103	35	36
		Effluent	4	8	19	13
	525 nm (m^−1^)	Influent	106	64	24	16
		Effluent	2	3	5	7
	620 nm (m^−1^)	Influent	76	48	18	5
		Effluent	1	1	3	1

**Table 3 membranes-12-00449-t003:** Peak position and functional groups of fouled and pristine membranes.

PT Peak Position (cm^−1^)	Probable Functional Group	MW Peak Position (cm^−1^)	Probable Functional Group
3500	1. N-H_2_ asymmetric stretching vibration of free NH_2_ 2. O-H stretching vibration of single bridged compound	3367	1. O-H stretching vibration of single bridged compound 2. N-H_2_ asymmetric stretching vibration of free NH_2_
3303	1. O-H stretching vibration of single bridged compound 2. N-H_2_ asymmetric stretching vibration of free NH_2_	2859	1. O-H stretching vibration of single bridged compound 2. N-H_2_ asymmetric stretching vibration of free NH_2_
2948	1. C-H asymmetric and symmetric stretching of alkane	2304	1. N-H_2_ asymmetric stretching of NH_2_ salt
2918	2. C-H asymmetric and symmetric stretching of alkane	806	1. N-H strong and broad stretching of primary and secondary amines 2. C-H strong stretching of aromatics 3. C-Cl medium stretching of alkyl halides
-	-	714	4. C-H rock medium stretching of alkanes

**Table 4 membranes-12-00449-t004:** Comparison of drinking water (Galeb d.d.) with ZW1-Pt effluent for washing dyeing machines.

Parameter	Drinking Water	Permeate ZW1-PT
pH	6.7	8.2
Total hardness (mg L^−1^ CaCO_3_)	214	67.7
Conductivity (µS cm^−1^)	545.0	1925
Magnesium (mg L^−1^)	7	5.87
Calcium (mg L^−1^)	64	17.5

## Data Availability

The data presented in this study are available on request from the corresponding author.

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
