# Peer review of "Investigation of Pretreatment of Textile Wastewater for Membrane Processes and Reuse for Washing Dyeing Machines"

_membranes, 2022, doi:10.3390/membranes12050449_

Round 1

Reviewer 1 Report

Agreed with the rebuttal of the authors.

Author Response

Thank you for your positive comment. English was additionally corrected.

Reviewer 2 Report

This manuscript deals with pretreatment selection for industrial scale separation of washing dyeing water. The study is actually quite simple and common, but because of the industrial scale, it can be accepted for publication. There are some issues to be corrected, as follows:

  1. Please make sure the supplementary figures and tables are uploaded.
  2. Line 15-21: Please merge this paragraph with line 9-14. Abstract is usually made of one paragraph only.
  3. Line 30-31: There is only one reference related to the textile wastewater treatment using membrane. Please add these references to widen the introduction.
  • J. Membr. Sci. 469 (2014) 306-315 https://doi.org/10.1016/j.memsci.2014.06.057
  • IOP Conf. Ser.: Earth Environ. Sci. 195 (2018) 012057 https://doi.org/10.1088/1755-1315/195/1/012057
  1. Line 112-114: The concentration, pH values, etc. come from what reference(s)? Please provide reference(s).
  2. Line 198: …attributed to the…
  3. Line 281-282: This is a short paragraph, please merge with the next one.
  4. Line 338: 3.5. Selecting the best pretreatment
  5. Line 424-447: This paragraph is too long. Please split to 2-3 shorter paragraphs.
  6. References: Please write the complete list of authors, do not use “et al.”
  7. Please check the journal’s template. Usually, the title of the articles is not written in italic. On the other hand. the journal name is usually written in italic.

Author Response

Q1: Please make sure the supplementary figures and tables are uploaded.

A1: Thank you for the comment and the comment was accepted. The supplementary figures and tables were be uploaded. It was our mistake for not uploading them.

Q2: Line 15-21: Please merge this paragraph with line 9-14. Abstract is usually made of one paragraph only.

A2: Thanks for the comment and paragraphs are merged. The change will be visible with track changes.

Q3: Line 30-31: There is only one reference related to the textile wastewater treatment using membrane. Please add these references to widen the introduction. . Membr. Sci. 469 (2014) 306-315 https://doi.org/10.1016/j.memsci.2014.06.057, IOP Conf. Ser.: Earth Environ. Sci. 195 (2018) 012057 https://doi.org/10.1088/1755-1315/195/1/012057

A3: Thanks for the comment. References are added. Number 8 and 9.

Q4: Line 112-114: The concentration, pH values, etc. come from what reference(s)? Please provide reference(s).

A3: Thanks for the comment but according to our experience reference is not necessary. The concentration and pH range are taken from experience and are different for each type of wastewater. Coagulation and coagulation/flocculation must be performed in wide range due to easier and more precise results.

Q5: Line 198: …attributed to the…

A5: Thanks for the comment. The change will be visible with track changes.

Q6: Line 281-282: This is a short paragraph, please merge with the next one.

A6: Thanks for the comment. The change will be visible with track changes.

Q7: Line 338: 3.5. Selecting the best pretreatment

A7: Thanks for the comment. The change will be visible with track changes.

Q8: Line 424-447: This paragraph is too long. Please split to 2-3 shorter paragraphs.

A8: Thanks for the comment. The change will be visible with track changes.

Q9: References: Please write the complete list of authors, do not use “et al.”

A9: Thanks for the comment. Wrong style was used and it was changed. The change will be visible in the references.

Q10: Please check the journal’s template. Usually, the title of the articles is not written in italic. On the other hand. the journal name is usually written in italic.

A10: Thanks for the comment. Wrong style was used and it was changed. The change will be visible in the references.

Round 2

Reviewer 2 Report

Review of membranes-1688612 v2

The authors have addressed the issues well, and provided sufficient argument. There are some minor corrections to be made, as follows:

  1. Line 188: FeCl3 --> with subscripted 3.
  2. Line 413: The name of the 4th author must be written as: de Souza Freitas, T.K.F. Be careful when writing surname with more than one word.
  3. Line 414: Dillenia indica --> in italic
  4. Line 442: Scientific Reports --> with uppercase R
  5. Line 450: Bulgarian Chemical Communications --> with uppercase C and C, respectively.
  6. Line 459: Environmental Science and Technology --> with uppercase S and T, respectively.
  7. Line 460-461: This is a strange title, and it even has “.pdf” in it. Please correct it, please rewrite the title in an appropriate way. Thanks.

This manuscript is a resubmission of an earlier submission. The following is a list of the peer review reports and author responses from that submission.

Round 1

Reviewer 1 Report

The manuscript probed the pretreatment of textile wastewater for membrane processes and the reusability of the treated water for washing dying machines. However, to me, the manuscript does not appear pinpointed towards its objectives. A lot of less important discussion has been made, especially in section 3. It would be better should the manuscript is made more specific. Nevertheless, I have the following observations on the manuscript.

Critical comments:

  1. The authors claimed that the MF is a better process for treating textile wastewater. They have highlighted the drawbacks of the techniques such as coagulation-precipitation, flocculation and adsorption. However, the authors have clearly shown that the MF process cannot withstand without the processes the authors have criticized already. How can it be concluded that the MF process is more efficient and cost-effective than the adsorption process? 
  2. In line 76: What was the reason for storing the wastewater at < 10 deg C?
  3. How were the FTIR samples made from the membrane surfaces? Did the membranes undergo sone drying process before sample preparation?
  4. The authors might also show FESEM images of the membranes at various stages of the study (pristine, fouling, washed).
  5. Line 123: To serve which purpose the membranes were stored in a dark and cold place?
  6. The authors have chosen to cite an article instead of giving some experimental details in two cases. The authors should include the experimental details at least in the SI file.
  7. Sec. 3.1 should be made more informative. The authors should provide the COD value, pH, EC, metal contents etc., along with the TOC and color of the wastewater. Further, the present form of Sec. 3.1 requires refurbishment. 
  8. The authors mentioned SF as a physicochemical process. What was chemical in the SF process? What kind of sands were used in the SF?
  9. The entire Sec. 3 contains too much discussion on the fundamentals of coagulation, coagulation/flocculation, etc., processes. This is not suitable for a research article. The excessive discussion should be trimmed out. 
  10. What volume of TWW produced 500 mL of sludge in the coagulation-flocculation process? What was the water content of the 500 mL-sludge?
  11. Sec. 3.2.4. also appears more like a literature review than a critical discussion on research outcome. 
  12. Line 257: How the initial flux decrease was estimated? This part requires some clarification. The initial flux decrease for all the studied membranes is not getting clear from Fig. 3. How the values 20, 25, 25, and 50 % were estimated?

Other comments:

  1. Line 25: The use of 'as' here is erroneous.
  2. From lines 25-31, some more references are required.
  3. Line 32: 'Pollutants content' should be pollutant-content.
  4. In lines 47-48, From where '436 nm' has come? What does it mean, actually?
  5. Where is the ANOVA for RSM studies? 
  6. Line 102: It would be better for the readers should the authors provide the full form of SF here again. 
  7. Line 103-104: Should be rewritten to bring clarity. 
  8. Line 292 requires rectification. 
  9. Line 325-329: Inconsistency in Tense has been observed. Rectification is required. 
  10. Line 338: Rectification is needed 

Author Response

Critical comments

Q1: The authors claimed that the MF is a better process for treating textile wastewater. They have highlighted the drawbacks of the techniques such as coagulation-precipitation, flocculation and adsorption. However, the authors have clearly shown that the MF process cannot withstand without the processes the authors have criticized already. How can it be concluded that the MF process is more efficient and cost-effective than the adsorption process? 

A1: Thanks for the comment. In the introduction it was stated that some scientists have shown that microfiltration is better than coagulation for the purpose of reducing COD and color. But the problem is the fouling of the membrane, which membrane separation processes are faced. This is not the case with coagulation and flocculation. Various examples of all possible TWW pretreatments are given what has been done so far in the area. The introduction part is changed and is more focused on the topic of the paper, i.e., on the reuse. Therefore, MF section is removed.

Q2: In line 76: What was the reason for storing the wastewater at < 10 deg C?

A2: Thanks for the comment. The reason for storage < 10 °C is that the physico - chemical characteristics of wastewater do not change too much. It would be best for the wastewater to freeze, i.e., be below 0 °C, but for now we do not have that option with huge amount of water, so we have to keep the water below 10 °C. Therefore, real textile wastewater (feed) was every time examined before each process.

Q3: How were the FTIR samples made from the membrane surfaces? Did the membranes undergo sone drying process before sample preparation?

A3: Thanks for the comment. In line 133 - 135 the following is visible: membranes were dried at 35 °C for 15 h in an oven after treatment to remove residual water.

Q4: The authors might also show FESEM images of the membranes at various stages of the study (pristine, fouling, washed).

A4: Thanks for the comment but at the moment this comment cannot be accepted. For now, we have attached FTIR spectra on which a change in membrane composition can be seen, i.e., membrane fouling. In future works, we definitely plan to record SEM as well. At the moment when this experiment was done we didn’t have possibilities to record SEM images.

Q5: Line 123: To serve which purpose the membranes were stored in a dark and cold place?

A5: Thanks for the comment. Membranes are stored in dark and cold place which means refrigerator. When membranes are bought, they are places in plastic bags and when samples of the membranes is taken we have to open the bag. Since bag is opened, we have to put in refrigerator to prevent microbiological fouling. We don’t have any other suitable place to preserve membranes and to prevent biological fouling.

Q6: The authors have chosen to cite an article instead of giving some experimental details in two cases. The authors should include the experimental details at least in the SI file.

A6: Thanks for the comment. The basic and important information’s about experimental part are written in the manuscript. Some additional information’s, according to reviewers’ comments are added and changes are visible. If authors understood reviewers’ comment, we cited articles where the apparatus are explained in detail. This is a policy of the journals since repetition if not recommended. Therefore, we gave a citation and all important procedures are written.

Q7: Sec. 3.1 should be made more informative. The authors should provide the COD value, pH, EC, metal contents etc., along with the TOC and color of the wastewater. Further, the present form of Sec. 3.1 requires refurbishment.

A7: Thanks for the comment. Authors agree that since work was performed with real wastewater more data about characteristics of wastewater if preferred but in this work focus was on this parameters since these parameters shows performances of the treatment and are most problematic. We don’t have ICP-MS so the content of the metals was not measured and according to our knowledge concentration of metal is not problematic in this water. Next reason is that the reuse was for washing dyeing machines, so all these parameters are not so important for it.

Q8: The authors mentioned SF as a physicochemical process. What was chemical in the SF process? What kind of sands were used in the SF?

A8: Thanks for the comment and the comment was accepted. There was an error in writing, SF is not a physical-chemical but a physical process and no chemicals were used in it. The change is visible in the manuscript. The type of the sand was added to section 2.2.

Q9: The entire Sec. 3 contains too much discussion on the fundamentals of coagulation, coagulation/flocculation, etc., processes. This is not suitable for a research article. The excessive discussion should be trimmed out. 

A9: Thanks for the comment and the comment was acceptable. The changes are visible in the manuscript in section 3.2.2, 3.2.3., and 3.2.4.

Q10: What volume of TWW produced 500 mL of sludge in the coagulation-flocculation process? What was the water content of the 500 mL-sludge?

A10: Thanks for the comment. Jar test was performed with 1 L of TWW which is mentioned in the section 2.2.. This is a standard procedure for the Jar tests. We did not measure the volume of water in 500 mL of sludge because we did not dehydrate the sludge and we don’t have centrifuge for this larger volume. It is known in the literature that the volume of sludge precipitated with iron salts is much less than 1/3 to 2/3 of the volume precipitated with aluminum salts.

Q11: Sec. 3.2.4. also appears more like a literature review than a critical discussion on research outcome. 

A11: Thanks for the comment and the comment was acceptable. The changes are visible in the manuscript in section 3.2.2 and 3.2.4. All theoretical part is removed from the manuscript and more references is added.

Q12: Line 257: How the initial flux decrease was estimated? This part requires some clarification. The initial flux decrease for all the studied membranes is not getting clear from Fig. 3. How the values 20, 25, 25, and 50 % were estimated?

A12: Thanks for the comment. Some text is added to clarify the initial flux decline. Hope it is better now. So, initial flux drop was calculated according to y-ax on the graph where it is written J/J0(water). J0(water), added in the manuscript, is flux of demineralized water and then flux during the treatment of TWW was divided by this flux. As can be seen if 1.0 is subtracted from 0.8 we get 0.2 which is a drop of 20%, etc.

Other comments:

Q1: Line 25: The use of 'as' here is erroneous.

A1: Thanks for the comment. The change is visible in the manuscript.

Q2: From lines 25-31, some more references are required.

A2: Thanks for the comment. Few new references are added.

Q3: Line 32: 'Pollutants content' should be pollutant-content.

A3: Thanks for the comment. Sentence was changed.

Q4: In lines 47-48, From where '436 nm' has come? What does it mean, actually?

A4: Thanks for the comment. 436 is SAC value at this wavelength but introduction is changed therefore this part is deleted.

Q5: Where is the ANOVA for RSM studies?

A5: Thanks for the comment. The suitability of the model is evaluated using variance analysis (ANOVA). Response surfaces are placed according to the most suitable model. Therefore, we do not consider it necessary to put models. This was used to obtained optimal conditions.

Q6: Line 102: It would be better for the readers should the authors provide the full form of SF here again.

A6: Thanks for the comment. The change is visible in the manuscript.

Q7: Line 103-104: Should be rewritten to bring clarity.

A7: Thanks for the comment. The change is visible in the manuscript.

Q8: Line 292 requires rectification.

A8: Thanks for the comment. The change is visible in the manuscript.

Q9: Line 325-329: Inconsistency in Tense has been observed. Rectification is required.

A9: Thanks for the comment. Step 1 changed.

Q10: Line 338: Rectification is needed.

A10: Thanks for the comment. Sentence is change and hope it is better now.

Reviewer 2 Report

In figures 1 and 2, the recovery can be plotted as separate figures for clarity.

Author Response

Q1: In figures 1 and 2, the recovery can be plotted as separate figures for clarity.

A1: Thanks for the comment and the comment was partially accepted. Authors believe reviewer means rejection. Figure 1 is changed and rejection is plotted separately than concentration and rejection for turbidity and TOC are in the same graph so it is easier to compare. Nevertheless, Figure 2 was not changed since authors think it is better to left on the same graph. Authors tried to separated values and rejection but could find good option, i.e., none of the option was not better.

Reviewer 3 Report

The introduction should be more focused and significance of the study should needs to be elaborated 

Which software you have used for the the statistical analysis?

For sand filtration process, you have not discussed the effects of TOC properly

The concrete discussion on the coagulation section needs support from the literature review 

In figure 3, the graph line is very thick and minor observations are not readable, please revise these type of graphs

The discussion of your results needs further support from the literature 

Author Response

Q1: The introduction should be more focused and significance of the study should needs to be elaborated

A1: Thanks for the comment and the comment was accepted. The changes are visible in the introduction part and we hope that the focus is now better.

Q2: Which software you have used for the statistical analysis?

A2: Thanks for the comment. Design Expert software. For the statistical analysis Box Behnken design was used and ANOVA for variance analysis.

Q3: For sand filtration process, you have not discussed the effects of TOC properly.

A3: Thanks for the comment and the comment was accepted. The changes are visible in the manuscript in the section 3.2.1.

Q4: The concrete discussion on the coagulation section needs support from the literature review 

A4: Thanks for the comment and the comment was accepted. The change is visible in the manuscript in the section 3.2.2.

Q5: In figure 3, the graph line is very thick and minor observations are not readable, please revise these type of graphs.

A5: Thanks for the comment and the comment was accepted. The change is visible on the graphs.

Q6: The discussion of your results needs further support from the literature

A6: Thanks for the comment but this comment is partially accepted. In a discussion part where it was considered necessary additional coverage with literature was added and is visible.

Round 2

Reviewer 1 Report

The authors have cleared many of the quries/comments.  It can now be accepted for publication.